# Graphical integrity issues in open access publications: Detection and patterns of proportional ink violations

**Han Zhuang**, **Tzu-Yang Huang**, **Daniel E. Acuna***

School of Information Studies, Syracuse University, Syracuse, New York, United States of America

* deacuna@syr.edu

## Abstract

Academic graphs are essential for communicating complex scientific ideas and results. To ensure that these graphs truthfully reflect underlying data and relationships, visualization researchers have proposed several principles to guide the graph creation process. However, the extent of violations of these principles in academic publications is unknown. In this work, we develop a deep learning-based method to accurately measure violations of the proportional ink principle (AUC = 0.917), which states that the size of shaded areas in graphs should be consistent with their corresponding quantities. We apply our method to analyze a large sample of bar charts contained in 300K figures from open access publications. Our results estimate that 5% of bar charts contain proportional ink violations. Further analysis reveals that these graphical integrity issues are significantly more prevalent in some research fields, such as psychology and computer science, and some regions of the globe. Additionally, we find no temporal and seniority trends in violations. Finally, apart from openly releasing our large annotated dataset and method, we discuss how computational research integrity could be part of peer-review and the publication processes.

**Data Availability Statement:** All relevant data and code are available in a Github repository (https://github.com/sciosci/graph_check) and Zenodo

## Author summary

Scientific figures are one of the most effective ways of conveying complex information to fellow scientists and the general public. While figures have the great power to leave a strong impression, they can also confuse and even mislead readers. Visualization researchers suggest several rules to avoid these issues. One such rule is the Principle of Proportional Ink (PPI) which suggests the size of shaded areas in graphs to be consistent with their corresponding quantities. The extent of violations of this principle in scientific literature is unknown, and methods for detecting it are lacking. In this article, we develop a deep learning-based method to perform such tasks on bar charts. An analysis of hundreds of thousands of figures revealed that around 5% of bar charts have violations of PPI. We found differences in the prevalence of these issues across fields and countries. We discuss the implications of these results and the opportunities and challenges posed by technology such as the one proposed here.

dataset (https://zenodo.org/record/5500684#.
YTvYclspBhF).

**Funding:** HZ, TYH, DEA were partially funded by
the ORI HHS grants ORIIR180041 and
ORIIIR190049. The funders had no role in study
design, data collection and analysis, decision to
publish, or preparation of the manuscript.

This is a *PLOS Computational Biology* Methods paper.

## 1. Introduction

Scientists need to understand previous research so as to build on others' work. This goal is best achieved when research findings are conveyed accurately and transparently. While inaccuracies can be the result of honest mistakes, some are also research integrity matters [1]. Some studies have estimated that 4% to 9% of scientists have been involved in research misconduct or have observed others' research misconduct [2,3]. However, assigning intentionality can be highly problematic, requiring expert opinion based on quantitative analysis. Regardless of intentionality, early, accurate, and scalable detection of potential problems during pre- and post-publication are crucial steps to make science more robust.

Publications contain a range of media, such as text, images, data, and graphs. Two of the most commonly studied media for integrity issues are text and images. Potential issues in text have revealed quality problems in English writing [4] and plagiarism [5]. Automated analysis of text is widely available, with several open, commercial, and free software to detect them [6]. Image-based media is more difficult to analyze because of the difficulty in parsing images. Studies driven by human examination have found that common manipulations of graphs and photographic images can have mistakes or integrity violations [7,8]. Some of these manipulations, such as adjusting the size and brightness of images, are common but might be misused and questionable. Several guidelines have been proposed to avoid these problems [9,10]. Still, recent advances in automatically detecting image issues offer auspicious results akin to text analysis [11,12]. Thus, text and images are commonly analyzed and have promising advances in automated analysis.

Scientific graphs tend to be the primary way of showing quantitative information in a visually-appealing manner. Compared to text and image integrity analysis, they are comparability much less studied. Still, graph issues can lead readers to misinterpret information [13], affecting their ability to make decisions [14]. Even if the readers are trained to detect problems, they can still be confused or misled [15]. Several general principles for graph construction have been proposed to avoid these issues [16]. One of these principles is the principle of proportional ink. It is a specialized rule derived from one of the general graphical integrity principles introduced by [16] who stated that "the representation of numbers, as physically measured on the surface of the graphic itself, should be directly proportional to the numerical quantities represented." Further, the derivation of the definition is succinctly presented by [17], stating that "when a shaded region is used to represent a numerical value, the size (i.e., area) of that shaded region should be directly proportional to the corresponding value." The violations of these guidances can be considered as graphical integrity issues but have been much less studied.

Researchers have proposed automated and scalable techniques to detect violations of research integrity and responsible research practices. The present study looks at another kind of scientific imagery that has been somewhat less explored before: graphs. To the best of our knowledge, we do not know the seriousness of graphical integrity issues among scientific graphs. Due to the nature of graphs, they are not shared in a machine-readable format, so it is not possible to apply methods such as those applied to text. Also, scientific graphs, such as line and scatter plots, have much less contrast than photographic figures, making it difficult to

apply computer vision techniques to extract information as these methods usually rely on detecting regions of interest defined by contrast differences. With the availability of new data-sets about scientific figures and open access publications, we have unprecedented opportunities to understand the extent of issues in graphs. However, we lack techniques and curated datasets to create methods to detect them effectively. While we cannot assume the intentionality behind the authors when a proportional ink principle violation is found, our work can act as a pre-filter for authors, journals, or parties interested in double-checking whether their graphical information presentation passes this basic filter of information visualization.

This study develops a novel method based on deep learning to detect graphical integrity issues automatically. We specifically focus on proportional ink principle violations of bar charts. In our study, a proportional ink violation is a y-axis that does not start from zero or has a truncation in its scale. Importantly, our framework is general enough that it can be adapted to other kinds of graphical integrity issues (e.g., data-ink principle). Armed with our technique (estimated to have an AUC of 0.917), we seek to answer how common these kinds of problems are in academia and whether there are systematic differences among researchers, journals, fields, and countries. Further, we examine whether graphical integrity issues "run in the family": does being a co-author of an article with issues in the past predict problems in the future? Finally, we discuss applications of these ideas beyond the proportional ink principle and graphical integrity domain.

## 2. Literature review

### 2.1. Graphical integrity

In visualization design, graphical integrity requires designers to create graphs reflecting the actual data. With this purpose, researchers have proposed several graphical integrity principles in the literature, and two of the most common are the principle of proportional ink and the principle of data-ink [16]. The principle of proportional ink was proposed by [17] and was inspired by Tufte, who stated that "the representation of numbers, as physically measured on the surface of the graphic itself, should be directly proportional to the numerical quantities represented" [16]. The principle of data-ink states that the graphs should use most of the ink to present the data instead of using it to show the background and aesthetic features. From the perspective of visual perception theory, the principle of proportional ink is based on how humans perceive the size of symbols in graphs [18] and the ratio of entities in graphs [19]. According to empirical visualization studies, violations of these principles can lead viewers to misunderstand the underlying data and results [13].

Researchers in business have extensively investigated graphical integrity in corporate reports and other forms of financial information, uncovering graph distortions and truncations of bar charts and other types of graphs [20,21]. Some researchers have examined the effects of these distortions and found they can greatly mislead viewers [22]. We would expect that these potential confusions would translate into scientific graphs as well.

Studies about graphical integrity issues in science are, in contrast, much less common. These issues, however, can be part of more significant problems such as misinformation [23], and therefore are essential to understand. Moreover, the degradation of information quality from science to the public might exacerbate the problems: researchers have found inadequate reporting of research in the news [24] and exaggerations of research from press releases [25].

### 2.2. Chart mining

In principle, researchers have the technology to investigate the graphical integrity of science automatically. One major step for evaluating graphic integrity automatically is to examine

graphs with chart mining techniques. Several studies have attempted to achieve this step through a broad field called chart mining [26]. For example, one study showed that we could tell the field of a manuscript by automatically analyzing its figures [27]. Another study developed machine learning techniques to extract information from graphs, including bar charts, line plots, and scatter plots [28–30]. For research integrity purposes, the key is to coordinate techniques from the field of chart mining and have large amounts of expertly annotated data. The curated data would allow machine learning models to learn what constitutes a violation of graphic integrity or not. We review some standard techniques in chart mining relevant to our research.

*Chart extraction*. Chart extraction is a relatively mature field that has enabled researchers to extract figures from scientific publications. Some of the chart extraction techniques exploit metadata available in PDFs or are based on computer vision algorithms that guess the location and size of figures within a document [31,32]. However, sometimes metadata might not be available, and therefore we can only rely on computer vision. With the development of neural networks, computer scientists have human-annotated documents to train deep convolutional neural networks to detect charts in documents without such metadata. The accuracy of these techniques has increased substantially over the years [33].

*Subfigure separation*. After chart extraction from publications, some figures might be compounded. Subfigure separation is the task that splits these compound figures into their subfigures [34]. Researchers have proposed several methods to achieve this separation, either using traditional computer vision algorithms such as edge-based detection of spaces between panels [35,36]. Interestingly, similar to what has happened with chart extraction, subfigure separation has benefited and improved from new deep learning techniques and large annotated compound figure datasets [37]. Although there is still room to improve, subfigure separation is mature enough to be used in production systems such as Semantic Scholar.

*Chart classification*. Different types of charts present different kinds of relationships. A bar chart might have a different way of expressing the proportional ink principle compared to a pie plot [16,38]. Therefore, we should ideally understand the kind of chart we are dealing with. The performance of this classification also has a cascading effect: the better we classify them, the more accurate our downstream analyses are. Researchers have used a range of techniques from traditional computer vision algorithms [39] to deep learning techniques [40] to perform this task. Currently, deep learning-based chart classification techniques have promising performance [34].

*Data extraction*. Data extraction is the most complex step as it attempts to reverse engineer the data used to produce the chart. A bar chart, plot, and pie chart panels might contain valuable information to be extracted from the image, such as point locations, tick values, and statistical information. It usually combines text detection, optical character recognition (OCR), keypoint analysis, and other advanced computer vision methods [41]. Neural networks have shown surprising performance at this task [42,43]. Depending on the type of graph, specialized data extraction techniques exist (e.g., bar charts or pie charts use different methods; see [44]). Data extraction is an active area of research that holds promising value for research integrity.

**2.2.1. Datasets to analyze science: open access publications and scientific imagery.** Chart mining techniques benefit from large datasets. Only recently, we have access to a much broader set of information pieces about science than ever before. These datasets include citations and other artifacts (e.g. [45,46]). Due to the push for open access publications, we also have unprecedented access to full text, citation contexts, figures, tables, and datasets [47]. These new datasets open up a wealth of opportunities to understand graphical integrity issues with data-driven techniques.

## 3. Results

### 3.1. Dataset annotation and method evaluation results

**3.1.1. Labeled dataset for evaluation.** To evaluate our method, we manually annotated a sample of bar charts from images of the large PubMed Central Open Access Subset (PMOAS) collection (see Materials and Methods). The PMOAS is part of the total collection of articles in PubMed Central. Our annotation pipeline starts with a random sample of 50,000 figures (see S1 Table). We then perform compound figure separation and bar chart detection (i.e., figure classification) (see Fig 1). The process detected 8,001 subfigures or figures as bar charts (Table 1). Then, two annotators (co-authors) manually annotated these bar charts as having proportional ink principle violations following the guidelines of [16,17]—more details in section 4.1.1. We found that 356 figures were classified as having integrity issues by at least one annotator. Our two human annotators had a high inter-rater agreement of 0.74 Cohen's Kappa [48]—a > 0.7 Cohen's Kappa is considered substantial agreement [49]. This high agreement among our annotators gives us confidence that there is a high potential to distill the rules that humans are using to classify violations.

Because our deep learning-based method needs to automatically extract features from the bar charts (e.g., tick labels, the start of the y-axis, etc.), we could only produce a reduced set of data from the 8,001 annotated bar charts. In particular, our method could only extract features from 4,834 of them (with integrity issue: 265, without integrity issue: 4569). This data was used for training the machine learning model in our deep learning-based method.

**3.1.2. Evaluation of the deep-learning-based method for estimating graphical integrity issues.** We used Area under the ROC curve (denoted as AUC), precision, and recall with stratified 5-fold cross-validation to evaluate the performance of our deep learning-based method (for our evaluation protocol and our method, see Materials and Methods section 4.2.3). AUC indicates the ability of our detector to classify two classes. Our average AUC across folds is 0.917, with a standard deviation of 0.02. In addition, the average precision across folds

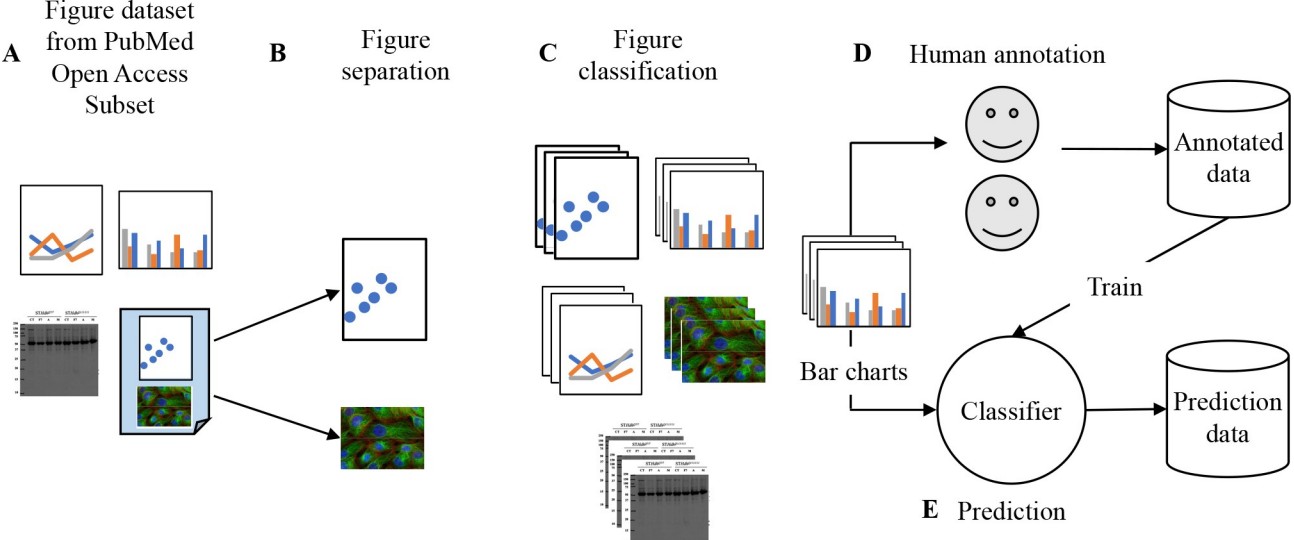

**Fig 1. Process for labeling and predicting violations of the proportional ink principle in bar charts.** A. Figures from the PubMed Open Access Subset (PMOAS) dataset are the source data. See Materials and Methods for details to (B) separate figures if they are compounded and (C) classify them into different kinds of charts. D. A large sample of bar charts are annotated by humans. E. These annotations are used to train a classifier and predict a large number of bar charts.

**Table 1. Summary of Training and Validation for Image Preprocessing.** [1][53], [2][30].

| Steps | Training data | Validation Data | Our Performance | Benchmark |
|---|---|---|---|---|
| Compound Figure Detection | ImageClef Medical 2016 Compound Figure Task | ImageClef Medical 2016 Compound Figure Task | Accuracy: 92% | Accuracy: 92% (Top one team in ImageClef Medical 2016) |
| Subfigure Separation | ImageClef Medical 2016 Compound Figure Task | ImageClef Medical 2016 Subfigure Separation Task | Score: 83% | Score: 84% (Top one team of ImageClef Medical 2016) |
| Chart Classification | Our generated charts | Revision data (not fully available) | Accuracy: 100% | Accuracy: 80%[1] |
| Text Localization | Localization from ArXiv papers through pdffigures | Our generated charts | F1: 76% | F1: 88%[2] |
| Text Recognition | No training data (Used an open-source fine-tuned model) | Our generated charts | F1: Exact 82% / Edit 90% | F1: Exact 95%/ Edit 98%[2] |
| Text Role Classification | No training data (Used an open-source fine-tuned model) | Our generated charts | F1: 80% | F1: 100%[2] |

is 0.77 with 0.0209 standard error. This result shows that around 77% of our positive predictions are true positive violations. The average recall across folds is 0.37 with 0.0207 standard error. In sum, these results reveal our method has its strength in making sure positive predictions are true while being relatively less successful with false negatives (see section 3.1.4 for error analysis).

**3.1.3. Representation quality of unlabeled data vs. labeled data.** Once we have our deep learning-based model trained, we set to analyze a large sample of data that is unlabeled. This is part of the prediction pipeline in Table 1. Before doing so, we want to make sure that the unlabeled data is similar to the labeled data in terms of their features. To do this, we compare the distribution of features described in section 4.1.2. Because we randomly sampled both datasets, we expected them to be similar. Indeed, we found that the unlabeled and labeled distributions are nearly identical (Fig 2).

## 3.2. Graphical integrity in open access journals

With this deep learning-based method, we can examine graphical integrity issues at scale. In the present study, we apply this method to open access journals to investigate the commonness and patterns of graphical integrity issues in science. In the following subsections, we first present a descriptive analysis of our data. Then, we offer analysis by countries, research fields, years, seniority of researchers, and the journal's impact. We present our analysis result for 250,000 predictions (prediction pipeline in Table 1) in the main text (S1 Text contains a similar analysis was done for the 50,000 human-labeled figures).

**3.2.1. Estimated prevalence of graphical integrity issues.** Although our annotated data contained 8,001 bar charts, only 4,834 of them were processable by our pipeline (S1 Table). This process involved computing the features for our classification model (see methods). The reason for this drop is that our automated pipeline was sometimes not able to extract the tick values or axes from the plots. Humans do not have such limitations, even at low resolutions.

In our human-annotated dataset, we found 265 proportional ink violations in the 4,834 annotated bar charts that could be processed (6.5% ± 0.4%). Per publication, this prevalence amounted to 5.5% (± 0.3%) of the publications with a bar chart. In our prediction pipeline, we predict that there are 479 proportional ink violations of the 20,666 bar charts that could be processed (2.3% (± 0.1%) per bar chart, 3.6% (± 0.2%) per publication with bar chart).

We found that the percentage of proportional ink violations is significantly larger in human-annotated than automated prediction bar charts (the *p*-value is less than 0.05 for both the ratio per publication with a bar chart and the ratio per chart by Fisher's exact test). Even though our deep learning-based method has achieved high accuracy, this difference suggests

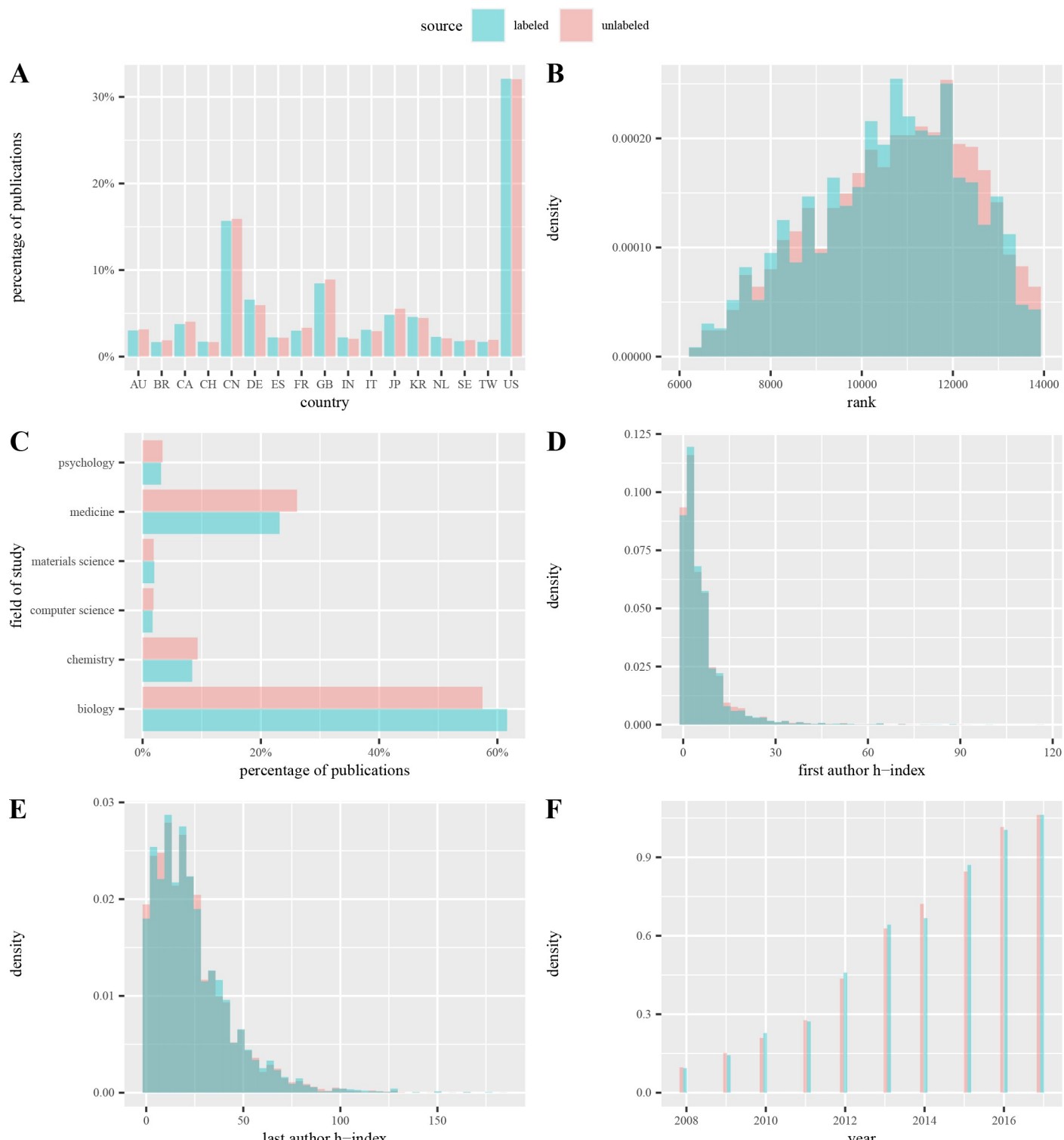

**Fig 2. Labeled vs. unlabeled data.** Features of both datasets are nearly identical. The prediction dataset could provide a reasonable estimation of the more general problems of graphical integrity.

that our method does not perform well on the metric of recall: we have a high number of false-negative cases. This is because some violations of the principle of proportional ink are hampered due to inaccurate text localization and recognition (see 4.2.1 section for more details). Also, given our conservative choice of threshold to decide whether a violation has occurred, we have this high false-positive rate in the performance test. We could change this threshold to bring our method closer to the human annotation data, but this inevitably increases the false positive rate. We discuss these issues in the Error Analysis section at the end of the result section.

**3.2.2. Differences by seniority.** We now investigate if there is a relationship between seniority and the ratio of graphical integrity issues (see Table 1 and Method and Materials section for more detail). To examine this research question, we conducted a correlation test between the likelihood of having graphical integrity issues and the *h*-index of the first author of publications and the last author of publications. (We acknowledge that the h-index of researchers might not fully reflect the seniority of researchers.) We found no statistically significant correlation for first author ($r = 0.0091$, $N = 9,078$, *p*-value = 0.38) or last author ($r = 0.18$, $N = 10,509$, *p*-value = 0.19). Similarly, we compared the h-index of the first authors with graphical integrity issues to those without and found no significant difference (Welch's t-test, $t(370.9) = 0.44$, *p*-value = 0.66). We did a similar analysis comparing the last authors and found no statistical significance (Welch's t-test, $t(408.58) = 0.76$, *p*-value = 0.45).

**3.2.3. Differences across the impact of journals.** High-impact journals are generally expected to publish rigorous research articles. Graphs as a part of research articles should also be accurate, but we do not know if the review of high-impact journals weeds out low-quality graphs. We tested the correlation between the rank of journals and journals' likelihood of having at least one article with graphical integrity issues. To measure this likelihood, we first compute the maximum likelihood that a publication contains a graphical integrity issue. Then, we compute the median of this value per journal. For the journal rank, we use the rank based on the importance of papers in the network of publications from Microsoft Academic Graph [46]. For example, a highly cited paper in a prestigious journal has more importance than a preprint article. A Pearson correlational analysis revealed that there is no correlation that the higher the ranking of the journal the more likely the journal has graphical integrity issues ($r = -0.01$, $N = 1357$, *p*-value = 0.63, Fig 3). This suggests that there is no relationship between journal rank and the likelihood of having integrity issues.

**3.2.4. Differences across research fields.** Graphical representations of information vary substantially across fields. To understand these differences, we compared the likelihood of having graphical integrity issues across research fields. The research field of journals is selected by researchers from Microsoft Academic Graph, and each publication is classified into research fields by a multi-class classification method with publications' content and connections in the network of publications. Different research fields have different commonness of graphical integrity issues ($F(5, 10137) = 22.18$, *p*-value < 0.001, Fig 4). A posthoc analysis (Tukey's HSD) reveals that psychology contains proportional ink violations significantly more frequently than chemistry (*p*-value < 0.001) and biology (*p*-value < 0.001). Also, computer science contains violations of proportional ink principles significantly more frequently than chemistry (*p*-value < 0.001) and biology (*p*-value < 0.001).

**3.2.5. Differences across countries.** Previous research has shown that there is a significantly different prevalence of research integrity across countries [8]. Here, we examined the likelihood of having articles with graphical integrity issues for all their authors from the country of their affiliations. We found that there is a statistical difference between authors from countries of their affiliation ($F(16, 11718) = 2.49$, *p*-value < 0.001). We further aggregate the

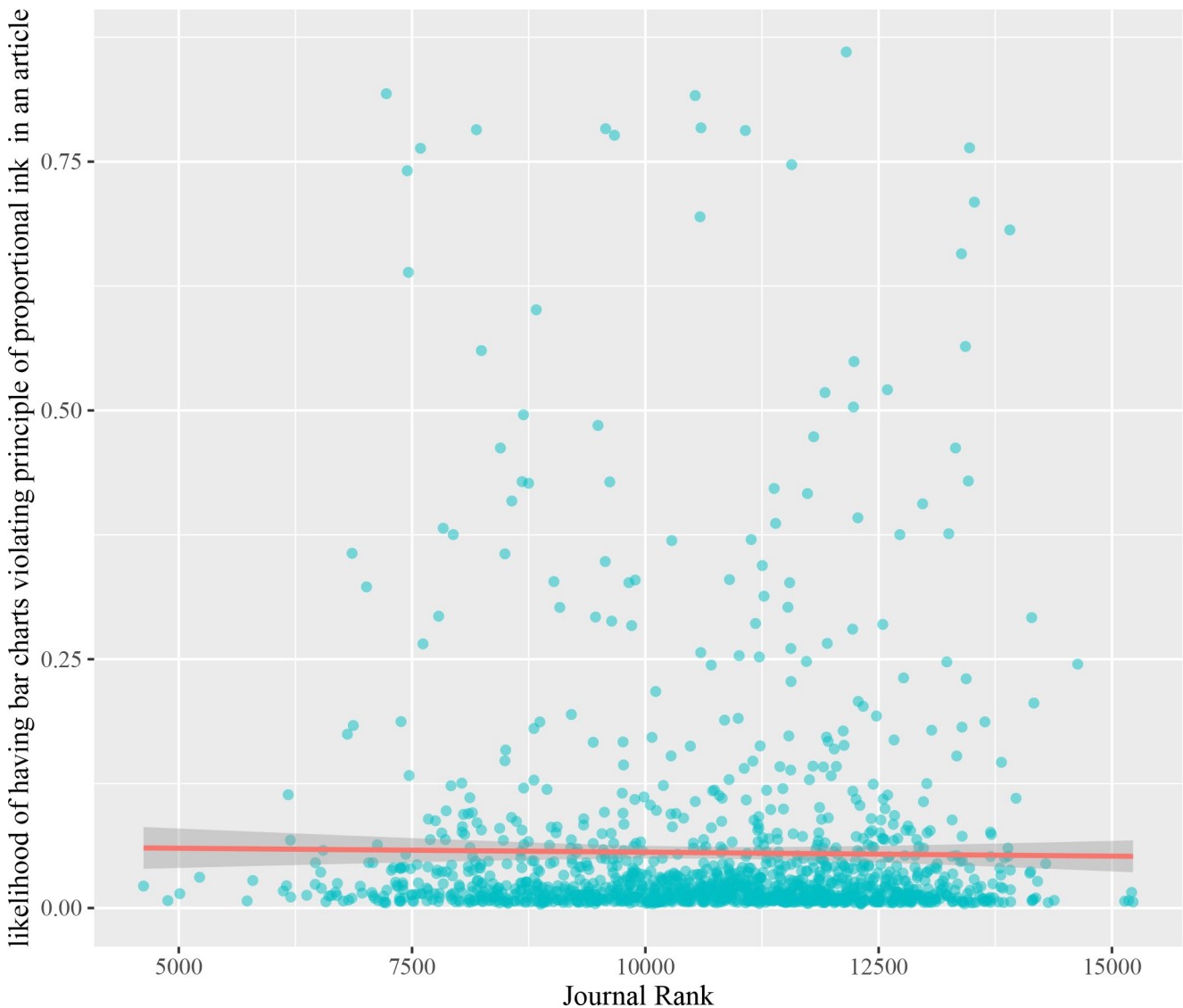

**Fig 3. Correlation between journal rank and the likelihood of having articles with graphical integrity issues.** No relationship between ranking and the median likelihood that an article has a bar chart violating the proportional ink principle.

likelihood of having graphical integrity issues per article for each country by taking the mean of these likelihoods (see Fig 5).

**3.2.6. Differences across years.** It is unknown whether publishers and research communities have been aware of graphical integrity issues in the past and taken actions to be protected from them. We computed the correlation between the likelihood of having graphical integrity issues and publication year to examine if graphical integrity issues are decreasing or increasing over time (see Fig 6). We did not find a relationship between these quantities suggesting that graphical integrity issues have been consistent ($r$ = -0.46, $N$ = 10, $p$-value = 0.18).

**3.2.7. Differences within author careers.** We now examined whether having experienced graphical integrity issues in the past is predictive of graphical integrity issues in the future. First, we examine if the graphs produced by authors who had graphical integrity issues in the past are more likely to have graphical integrity issues than the graphs produced by authors

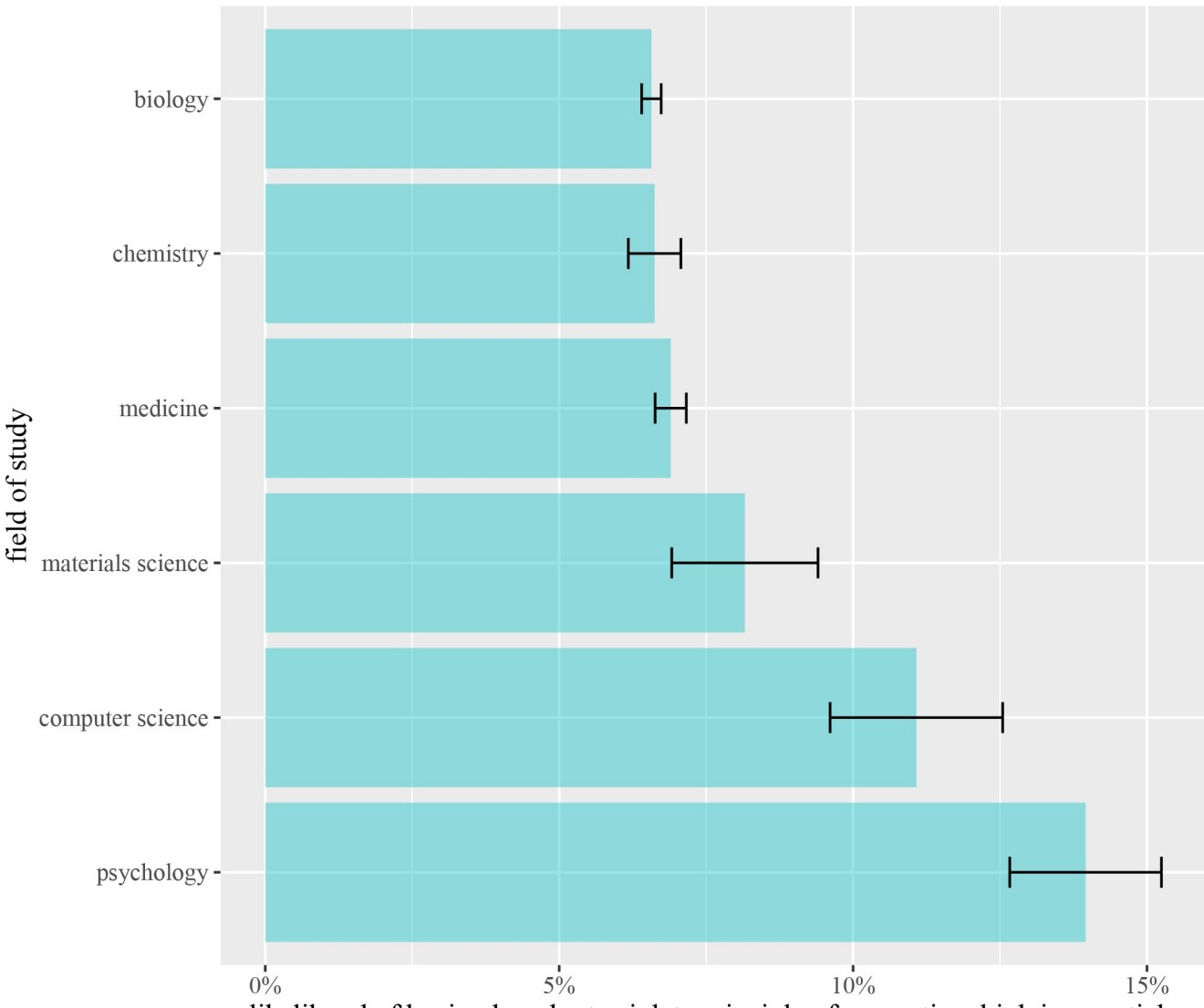

**Fig 4. The likelihood of having graphical integrity issues across each research field.**

who were not found with graphical integrity issues. We analyzed 301 bar charts in the future and found that only 5 of them had problems. Of the graphs produced by authors who did not likely have graphical integrity issues in the past, we analyzed 568 bar chart features and found that 6 of them had problems. According to a Fisher's Exact Test, the difference between these two groups is not statistically significant (*p*-value = 0.53). Second, we then analyzed it at the author level. We found that 7 out of 218 authors with issues in the past had problems in the future; we found that 10 of 423 authors with potential issues had problems in the future. Again, Fisher's Exact Test's difference between these groups of authors is not statistically significant (*p*-value = 0.61). These results suggest that graphical integrity issues do not seem to "run in the family": having issues in the past might not make an author have problems in the future.

**3.2.8. Error analysis.** Interpreting the differences between human annotation and our method improves the transparency of our method, but this task is difficult. To interpret this

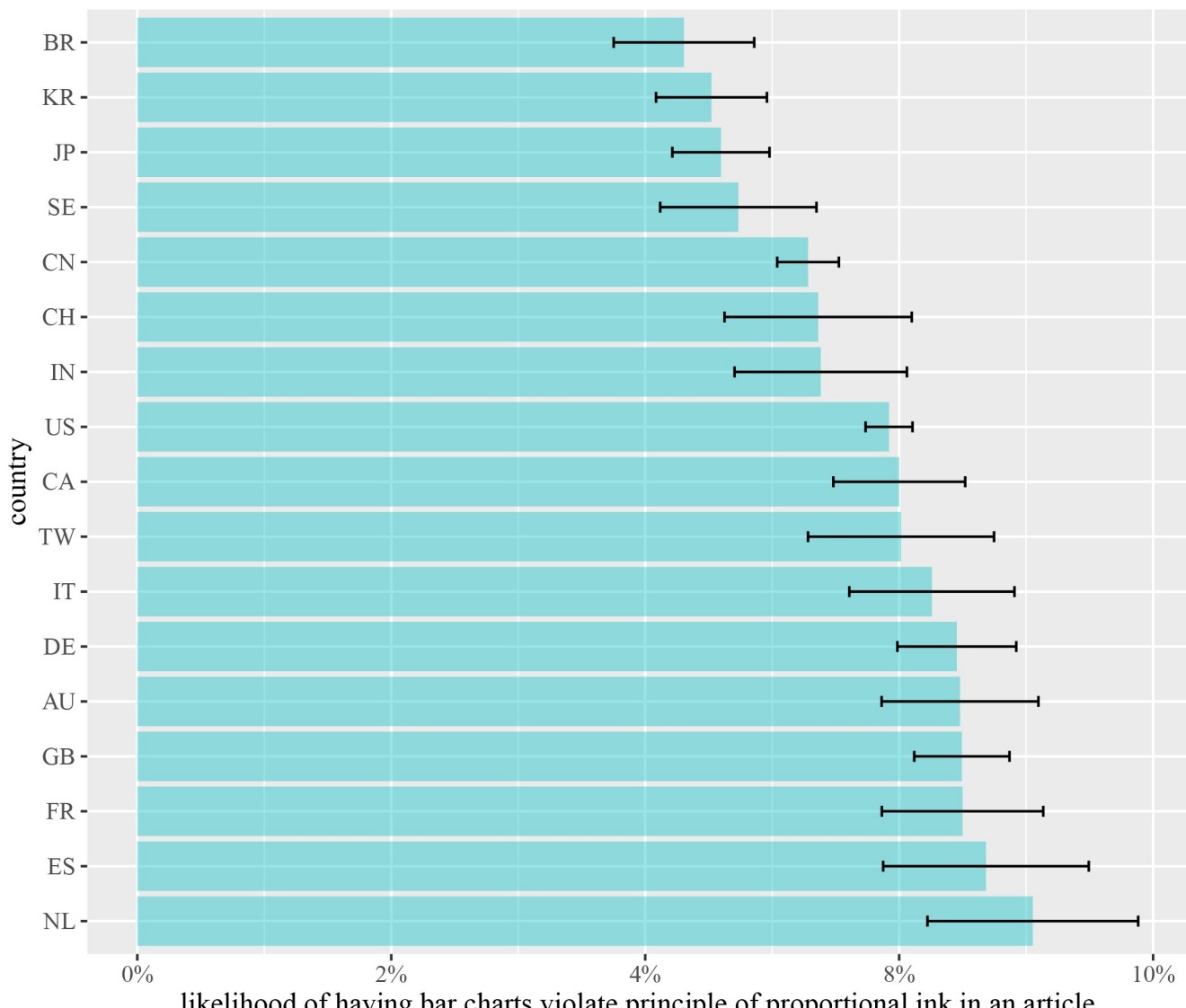

**Fig 5. The likelihood of having graphical integrity issues across each country.** Top three countries as the Netherlands, Spain, and France.

difference, we conducted a systematic evaluation of how the parameters on our model change the precision, recall, and predicted prevalence in the datasets. This is because we want to examine why our human annotators find 5.5% of bar charts having proportional ink violation, but our deep learning-based method finds 2.3% of bar charts having proportional ink violation (see 3.2.2 section for more details). We found this discrepancy between predicted *prevalence* and the human-annotated *prevalence* can be the result of the threshold in our classifier for classifying a bar chart into proportional ink violation (see section 4.2.3 for more details). We can reduce the discrepancy between predicted prevalence and human-annotated prevalence by lowering this threshold. Although this comes at the cost of increasing the false positive rate, we can approximate the human and predicted prevalence rate with a reasonable cost. To be concrete, we explore parameters that would approximate human prevalence. When we change the default threshold of classifying bar charts to graphical integrity issues from 0.5 to 0.23, we can improve our predicted prevalence rate of graphical integrity issues from 2.3% to 3.4%, without

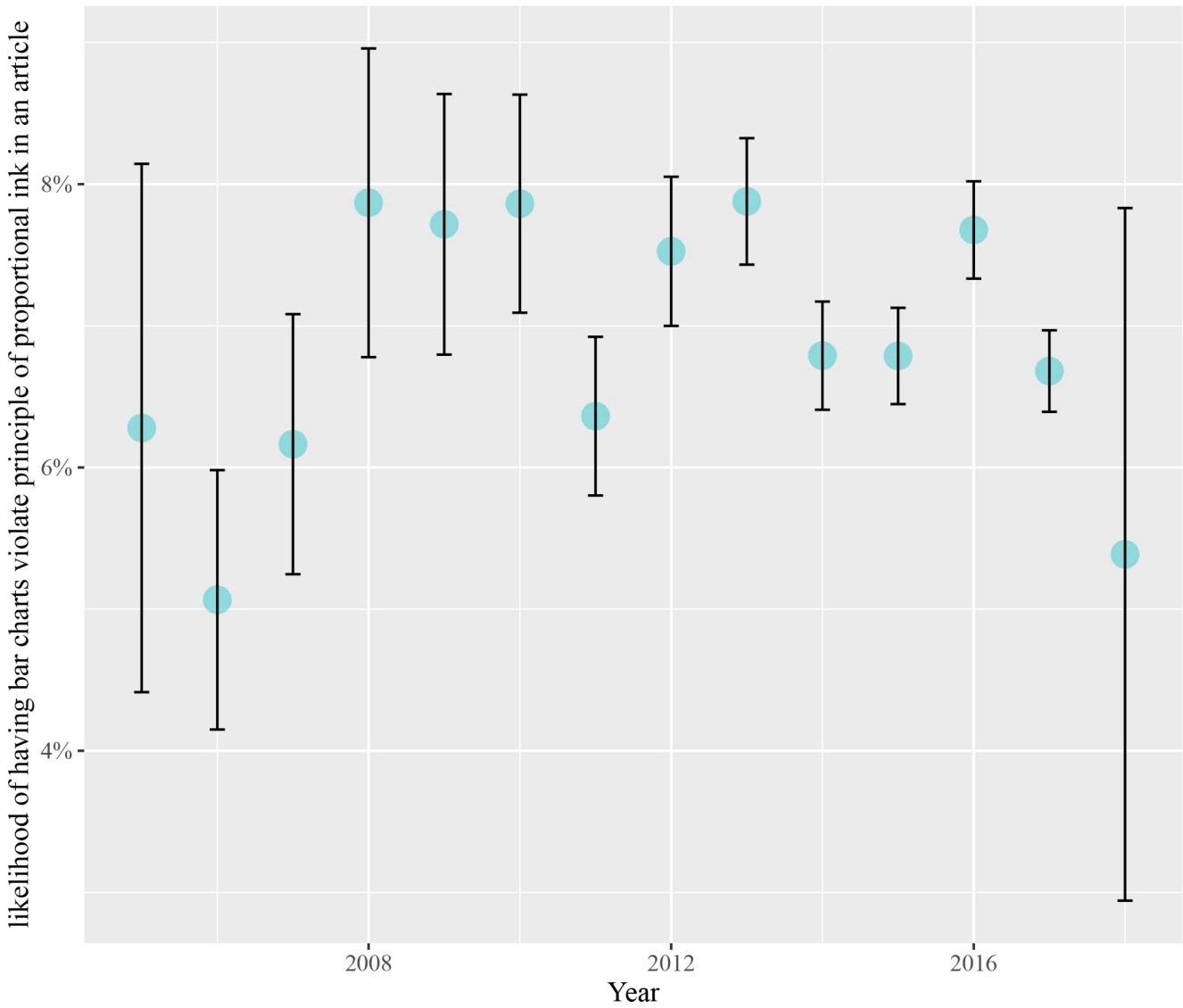

**Fig 6. The likelihood of having graphical integrity issues across each year.**

losing the F1 score (see S3 Fig). Thus, depending on the ultimate goals of the system, we do not necessarily want to approximate the human prevalence rate in this fashion. We believe that the choice of such a threshold should be in the hands of the users of the systems—ultimately, authors, journals, and universities.

We also performed a systematic analysis of the false positive and false negative classifications to determine whether we need more data, more sophisticated methods, or better feature engineering (see S4 and S5 Figs). The overall takeaway from this analysis is that our method has issues detecting bar charts with a partially hidden y-axis. This is because, for our method, we need highly accurate text localization and text recognition on the y-axis to perform well on the detection of these violations. However, methods for text localization are an area of active research [50].

### 3.3. Regression analysis

All the previous analyses were correlations between the variable of interests (e.g., author's seniority) and the outcome (i.e., likelihood of proportional ink violations). However, we should also account for the partial correlation among the independent variables before relating it to the outcome. To this end, we run a multivariate regression analysis relating journal ranking, research field, citation count, author seniority (h-index), year of publication, number of affiliations, and country of affiliation to the likelihood of proportional ink violation. The model explains a statistically significant and weak proportion of variance ($R$ squared = 0.02, $F(42, 8707) = 4.51$, $p < .001$, adjusted $R$ squared = 0.02). The details of the regression are available in the S2 Table. Our results found that higher relationship between some fields (computer science: standardized coefficient 0.036 (standard error = 0.011), $t(8707) = 3.41$, $p < 0.001$ and psychology: standardized coefficient 0.11 (standard error = 0.0084), $t(8707) = 9.75$, $p < 0.001$) and lower relationship between some countries (e.g., Japan: standardized coefficient -0.037 (standard error = 0.0070), $t(8707) = -3.33$, $p < 0.001$; Korea: standardized coefficient -0.03 (standard error = 0.0073), $t(8707) = -2.57$, $p = 0.010$) to produce proportional ink violations. Interestingly our regression shows that journal rank has a significant positive effect on the likelihood of having proportional ink violation (standardized coefficient -0.048 (standard error < 0.001), $t(8707) = -2.75$, $p < 0.001$). This journal rank effect implies that higher-ranked journals are more likely to contain violations of proportional ink. This effect is there even after controlling for the size of the journal (i.e., number of articles published) and the interaction between ranking and size of the journal.

## 4. Materials and methods

### 4.1. Materials

**4.1.1. PubMed open access and annotations.** PubMed Open Access is a subset of PubMed Central, consisting of thousands of journals in biomedical research [47]. To train our machine learning-based graphical integrity evaluator, we annotated one subset of collected images mentioned in section 3.1.1 (after compound figure detection, subfigure separation, and chart classification) as graphical integrity issues and others (see Fig 7; also see S1 Table). We consider images as graphical integrity issues if they violate the principle of proportional ink. The specific rules are: a bar chart's y-axis should start from zero, have one scale, and not be partially hidden [16,17]. We annotate images as *others* if they follow all three rules, do not have a full y-axis or x-axis, or are not bar charts.

**4.1.2. Datasets for image processing techniques and features for statistical analysis.** To automate graphical integrity evaluation, we need image processing techniques to obtain information from images. This study has six image processing steps: compound figure classification, subfigure separation, image classification, text localization, text recognition, and text role prediction (see the summary of training data and validation data in Table 1).

*Data for testing within-author integrity issues*. After labeling images, we analyzed the publications of these authors in the following three years after the first year we annotated. Based on the labeled data, we found 218 authors who produced graphs with integrity issues in their publications and analyzed 301 bar charts in their following publications. Also, there are 4780 authors who have not been found graphical integrity issues in our labeled data. Due to the unbalanced amount between the two groups, we randomly selected 3000 images from the following publications and analyzed 568 bar charts.

We now describe the features used in our statistical analyses:

*Seniority of researchers*. As the research experience of researchers grows, some researchers might be aware of graphical integrity and want to avoid graphical integrity issues. Thus,

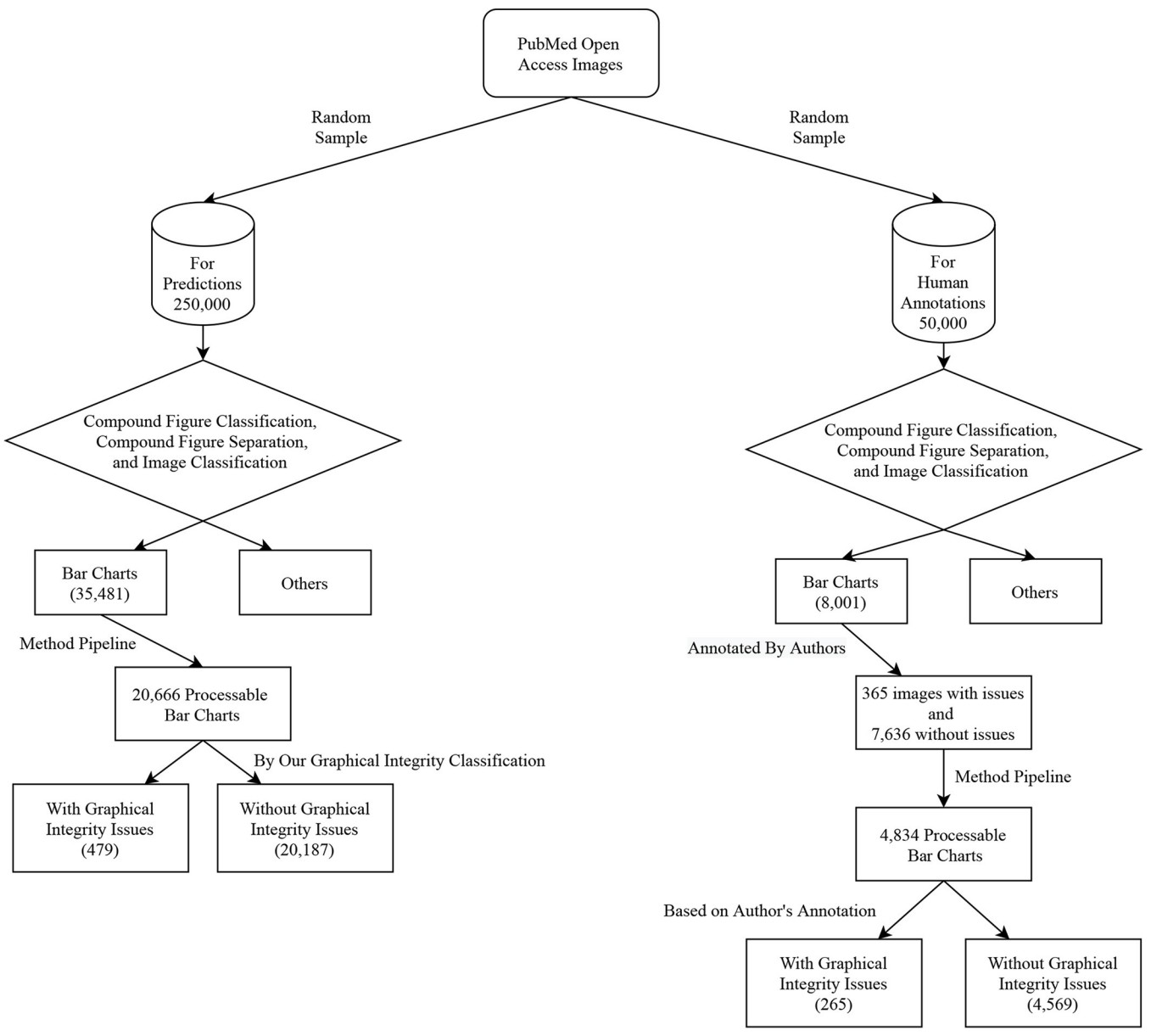

**Fig 7. Flowchart of our data source and process.** Predictions and Human Annotations data sets are randomly selected from PubMed Open Access Images. Authors annotated 8,001 bar charts from the human-annotated set, and 4,834 bar charts could be processed by the method pipeline.

graphic integrity issues might be less common among senior researchers, for they have more education and knowledge. If this hypothesis holds, then it might imply that researchers can learn graphical integrity over time by themselves. If not, research communities might need more education about this topic to reduce such problematic practices. To examine this hypothesis, we collected researchers' h-index to proxy their seniority for the analysis.

*Impact of journals.* High-impact journals usually have a strict standard for their publications. It is natural to expect high-impact journals to publish more rigorous research findings. However, whether high-impact journals value graphic integrity is unknown. Thorough research should make sure its visualizations convey accurate information to readers. If not, the

journal publishers might need to include graphical integrity into their review standard. To examine this research question, we collected the rank of journals of articles in our sample.

*Research fields.* Different disciplines have their traditions, such as citation format and writing style. Similarly, other disciplines might have additional requirements for graphical integrity. However, this is a new and important research question because the publishers might want to enhance the review of graphical integrity for specific research fields. To address this question, we collected the field of study of journals of articles in our sample.

*Countries.* Education and culture vary from country to country. But we do not know if some countries have stricter graphic integrity. This research question can help researchers to know which country needs more protection of graphical integrity. Thus, we collected the country of the author affiliation of articles in our sample.

*Year of publication.* Science is changing with our research policy, education, and society over time. The time-series analysis of science helps researchers to understand the direction of new science. Similarly, graphical integrity over time can help the public to know if the awareness of graphical integrity changed in the past. To answer this research question, we collected the year of publications of articles in our sample.

## 4.2. Methods

Fig 8 explains the steps of our method, and Fig 9 shows an example of how our method analyze images.

**4.2.1. Image preprocessing.** The goal of image processing is to obtain information from images for feature engineering and classification of graphical integrity issues. Our image preprocessing includes the following steps:

*Compound figure classification.* Compound figures are standard in academic papers (e.g. [27]), but usually, graphical integrity issues are defined on non-compound figures. Therefore, we use a high-quality feature extractor based on a convolutional neural network (Resnet-152 v2, pre-trained on ImageNet) to classify figures into compound figures and non-compound figures.

*Compound figure separation.* In the case of a compound figure, we need to separate compound figures into subfigures because subfigures in compound figures might also have graphical integrity issues. To achieve subfigure separation, we trained a convolutional neural network (YOLO v4, pre-trained on MS COCO dataset) to localize subfigures in compound figures [51].

*Image classification.* This study focuses on bar charts because the violation of the principle of proportional ink is a widely-accepted graphical integrity issue for bar charts. Thus, we generated charts (bar charts, line charts, scatter charts, heatmap charts, box charts, area charts, radar plots, maps, pie charts, tables, Pareto charts, Venn diagrams, and violin charts) and collected diagnostic figures from ImageCLEF 2016 competition [34]. Then we use a high-quality feature extractor based on a convolutional neural network (Resnet-152 v2, pre-trained on ImageNet) to classify figures into categories.

*Text localization.* To predict if a bar chart has graphical integrity issues, we also need to know the texts on images for our classification. Therefore, we fine-tuned a convolutional neural network (YOLO v4, pre-trained on MS COCO dataset) to detect or localize texts on academic figures, preprocessed with Stroke Width Transformation [51,52].

*Text recognition.* More than the location of texts on graphs, we also need their content. Thus, we used Tesseract to recognize the content of texts based on the predicted locations of texts from text localization. We used one fine-tuned Tesseract model of English texts for this task.

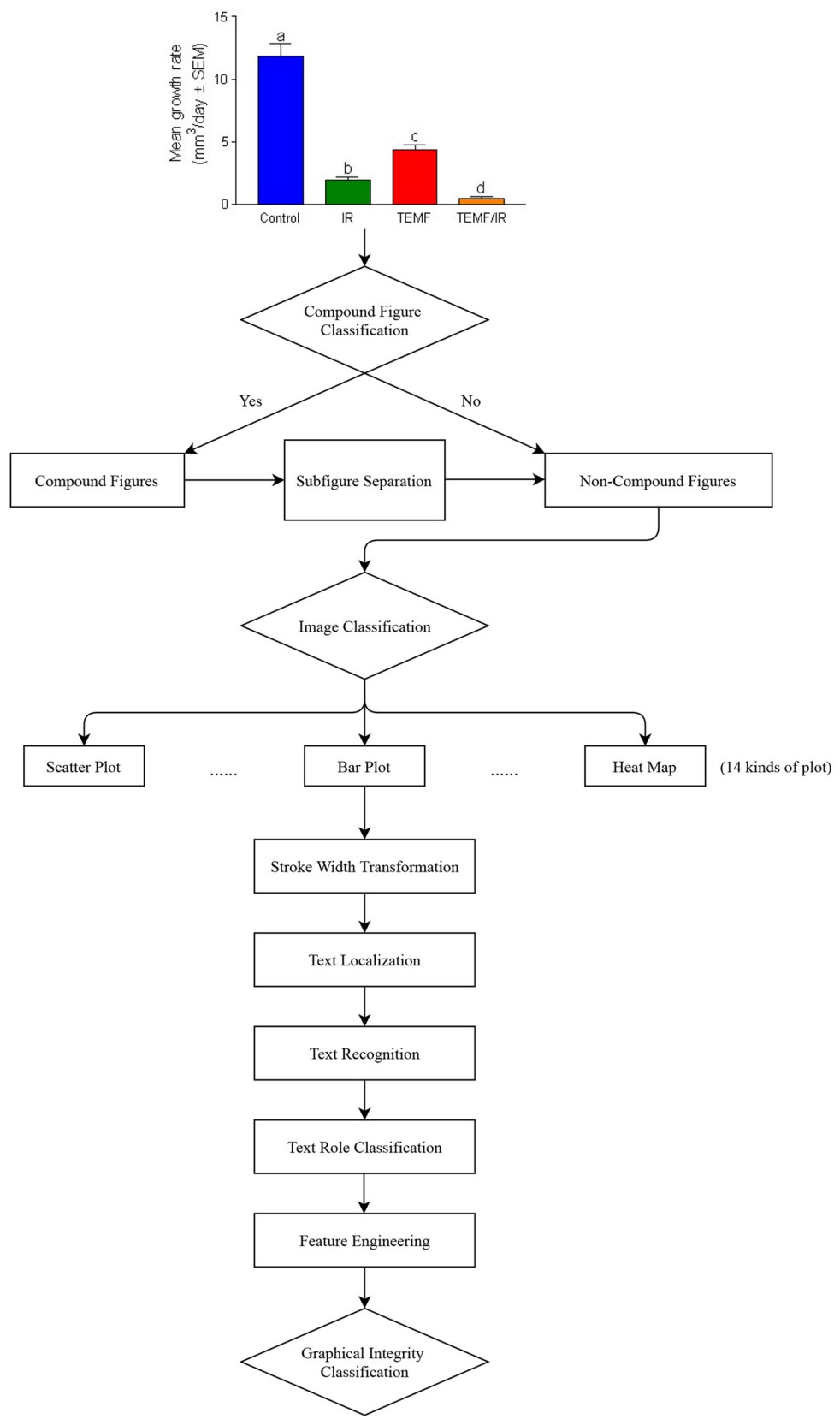

**Fig 8. Flowchart of our graphical integrity classification.** Refer to Fig 9 for comparison and Section 4.2 for method details of each step in our deep learning-based method. In our method, we apply compound figure classification, subfigure separation, text localization, text recognition, text role classification, and graphical integrity classification. To reproduce our method, see our code in https://github.com/sciosci/graph_check.

*Text role classification*. Also, we need to know the role of texts on each figure for our analysis and thus use one open-source model to predict the role of texts on academic figures, based on the geometric information of texts on graphs [30].

*Image preprocessing validation*. To evaluate the performance of previous tasks, we used public datasets and our own generated datasets (when we could not find public datasets). The summary of our performance and benchmarks is summarized in Table 1.

**4.2.2. Feature engineering.** One way to build an automatic graphic integrity issues classifier is to apply classifiers from machine learning. A machine learning-based classifier needs features from data instead of the raw data. Thus, we conducted a feature engineering step from our image preprocessing (Table 2). Because the principle of proportional ink requires the ratios between the numeric value and the area of each bar to be the same in the same bar chart, the violation of this principle has the following symptoms for academic figures: bar charts with a non-zero y-axis and bar charts with a y-axis of multiple scales (also considering partially hidden y-axis). These two kinds of bar charts all violate the principle of proportional ink. Although bar charts with a log scale also violate the principle of proportional ink, we do not consider these bar charts as graphical integrity issues because log-scale is widely accepted in academic figures [54]. To examine if one bar chart has the above symptoms of graphical integrity issues, we transform the information from our image preprocessing into features for the final classification. The first two features are derived directly from the principle of proportional ink for bar charts. However, based on our experiments, these two features are not enough to detect graphical integrity issues because our image preprocessing might introduce errors to these two features. For example, the y-axis label might not be recognized by our text localization model. Thus, we created other features to consider the errors from our image preprocessing.

**4.2.3. Proportional ink principle evaluator.** For an automatic graphical issue detector, rule-based methods are feasible when the features are highly correct. However, given the errors introduced from image processing, the texts and their roles on figures might not be correct. Thus, we took a probabilistic approach: the random forest model from machine learning. Our random forest model can predict if a figure is misleading or not based on the features of figures from our image processing techniques (see Fig 8).

**4.2.4. Performance evaluation method.** We used three metrics to evaluate our method: AUC, precision, and recall. Before we introduce how to interpret AUC, we first explain how we define true data points, false data points, positive data points, and negative data points, true positive data points, false-positive data points, true negative data points, and false negative data points. Here, true data points are images, which have an annotation as a proportional ink violation. False data points are images, which have an annotation as no proportional ink violation. Positive data points are images, which have a prediction from our method as a proportional ink violation. Negative data points are images, which have a prediction from our method as no proportional ink violation. True positive data points are images that have a prediction from our method as a proportional ink violation and also have an annotation as a proportional ink violation. False-positive data points have a prediction from our method as a proportional ink violation but have an annotation as no proportional ink violation. True negative data points are images, which have a prediction from our method as no proportional ink violation and annotation as no proportional ink violation. In addition, false-negative data

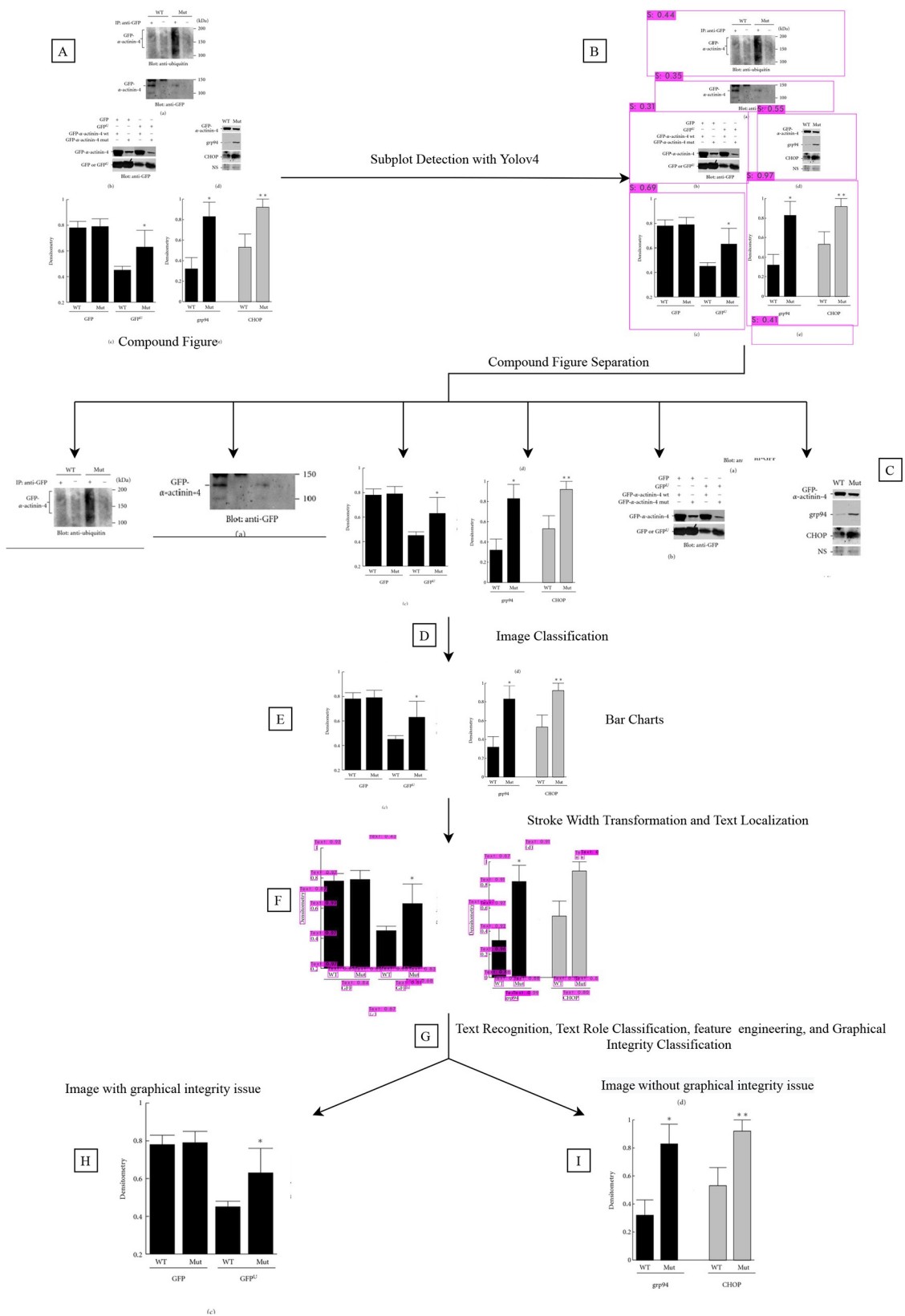

**Fig 9. An example process for predicting violations of the proportional ink principle (see Materials and Methods for details, and our code is in https://github.com/scioSci/graph_check). A.** Input image representing a scientific figure. PubMed Open Access subset provides figures already extracted from the publications. **B.** Subplot extraction using the YOLO deep learning architecture [51] trained on the hand-annotated dataset (see Materials and Methods). **C.** Each subplot is extracted from the input image. **D.** Subfigure plot classification where only bar charts are extracted (E). For each bar chart, we detect a set of low-level features (F), which are later used for predicting whether a bar chart is violating the proportional ink principle (H, yes) or not (I, not).

points are images, which have a prediction from our method as no proportional ink violation but have an annotation of proportional ink violation.

AUC is the area under the ROC curve, which shows how true positive rate (the ratio between the number of true positive data points and the number of positive data points) changes with the false positive rate (the ratio between false-positive data points and the number of negative data points) with various thresholds as a curve [55]. When we have a high AUC for classifiers (close to 1), the true positive rate is high, and the false positive rate is low, and we have a much more accurate classifier than doing classification in a random manner. When we have an AUC of 0.5, the classifier is similar to doing classification in a random manner.

We additionally investigated the precision and recall of our method. Precision is computed as Eq (1), which is the ratio of the number of true positive data points to the number of all positive data points [56]. The recall is the ratio of the number of true positive data points to the number of all true data points (Eq (2)).

$$Precision = \frac{True\ Positive}{True\ Positive + False\ Positive} \tag{1}$$

$$Recall = \frac{True\ Positive}{True\ Positive + False\ Negative} \tag{2}$$

## 5. Discussion

In this article, we attempt to understand the extent of graphical integrity issues in scientific publications. We focused on violations of the proportional ink principle, one of the most basic visualization rules. We developed an automated method based on deep learning to detect these issues by learning from a large sample of human-annotated bar charts. Our approach achieved a high accuracy (AUC = 0.917). Our results suggest varying degrees of prevalence across research fields, journal rank, and affiliation countries. We now also discuss contributions of our method, implications of our research findings, limitations of our study, and future work.

**Table 2. Summary of features for proportional ink violation detection.**

| No | Feature Description | Reason |
|---|---|---|
| 1 | The value of the lowest y-axis label on the y-axis (detected or inference from y-axis) | The lowest y-axis label should be zero |
| 2 | The increasing rate between each pair of y-axis labels | The scales of y-axis should be consistent across each pair of neighbor y-axis labels |
| 3 | If we need to inference the lowest text on the y-axis | If the lowest label on the y-axis is far from the x-axis, then we might ignore the actual lowest label on the y-axis |
| 4 | If the y-axis has a mix of integer and float number | Tesseract might not perform well with float number, and thus the increasing rate in the y-axis might not be accurate |
| 5 | The probability of being texts | We prefer texts with a higher probability of being texts |
| 6 | The OCR confidences of texts on the y-axis | We prefer predictions of the content of texts with a higher confidence |
| 7 | The probability of being bar charts | Our classifier only classifies bar charts. Thus, we prefer figures with a high probability of being bar charts |

### 5.1. Contributions

This study contributes an automated method to graphical integrity investigations and chart mining. Our method can detect the violation of the principle of proportional ink automatically and shows the potential to detect other graphical integrity issues if we follow the same steps. In previous studies, researchers have relied on human experts on graphical integrity investigations [21,22]. Although human experts can detect graphical integrity issues with very high accuracy, the speed of annotation can be slow. For a large sample of graphs, we still need automated techniques for graphical integrity investigations. Trying to fill this gap, we proposed our automated method to examine images at scale and to detect proportional ink violations with relatively high accuracy. Moreover, the components in this method can address compound figure classification and subfigure separation for other graphical integrity investigations, complementing previous research (e.g., [35,37]).

Our work shows the potential of using AI to help publishers check integrity issues. Given the overwhelming volume of new manuscript submissions, editors and reviewers face the challenges of maintaining the quality of publications. One potential way of reducing the workload of editors and reviewers is to apply AI to conduct some basic checks on manuscripts. With such considerations, one utility of this method is to analyze a large collection of graphs to detect the violations of the principle of proportional ink. Also, our method provides compound figure classification, subfigure separation, chart classification for publishers to investigate graphical integrity, image reuse, or image fabrications. Publishers could use these components for displaying and further processing purposes.

### 5.2. Implications

Our study found a varying prevalence of proportional ink violations across research fields and affiliation countries. Our result implies that graphical integrity issues might be more common in specific groups of researchers and various research communities. From the aspect of research fields and graphical integrity, our result shows that psychology and computer science might have more graphical integrity issues.

Our country and journal differences differ from image integrity studies (e.g., [8]). For example, we found that European countries tend to have a higher likelihood of graphical integrity violations. The difference in proportional ink violation across fields might be the result of the coverage of research regulation across research fields. For example, some fields have better research integrity regulations than others [57].

The high accuracy of our method implies that it should be viable to be applied before or after peer review. Journals and reviewers are already overwhelmed enough that checks like our methods are likely to be overlooked. More broadly, our results show that modern computational research integrity technology has the great potential to scan for potential issues at scale. If our analysis is integrated with other kinds of analysis for text and image plagiarism, the scientific publishing system could offer a good chance to improve the overall state of the quality of scientific communication.

### 5.3. Limitations and risks

*Method limitation*: Our proposed method makes mistakes in compound figure classification, subfigure separation, and bar chart classification (see Table 1). Thus, some bar charts might not be separated from compound figures correctly or might not be classified into bar charts at all. The method depends on accurate text location and parsing (see Table 1). Future work will incorporate more recent developments in deep learning technology for the detection, location, and parsing of text in images.

Our analysis focuses on proportional ink violations among bar charts only. However, bar charts are only one of many types of charts used in scientific publications (see Table 1). Future studies will adapt our method to detect these kinds of charts and detect adherence to graphical integrity principles.

*Generalizability of research findings*: Our method cannot detect all graphical integrity issues, and perhaps we are underestimating their prevalence. Our analysis might not be generalizable to non-open access journals. Non-open access journals tend to have a higher impact than open access journals [58] and can have different review processes. In the future, analyzing PDFs and figures of traditional journals would be an obvious next step.

*Risks*: Our method might be "gamed" by some users to pass our checks. It is unclear to what extent this happens with text plagiarism. We can think of two extreme potential responses to this. The first response is to simply let both good actors (e.g., research integrity investigators, editors) and bad actors (e.g., unscrupulous parties) use our method freely. We see this approach as similar to the one that search engines take: people can game search engine algorithms, but we will adapt. In this sense, we should constantly be paying attention to the bad actors in a game of cat and mouse. According to our analysis, the extend of the problem is worrying (5% have violations), but releasing the tool openly can help spur search on the topic. We believe this approach might be the right cost-benefit response to this kind of research integrity issue.

A second answer to the issue is to keep our method secret and only give access to it to publishers and reliable parties. This would make the process less widely available, but it would prevent scientists who genuinely want to improve their manuscripts to not being able to access this technology. We think this is a reasonable strategy, but the entity or researcher giving access to this method should be responsive enough to not stifle innovation in this space.

## 5.4. Future of our work

In the future, we hope to incorporate this analysis with other kinds of graph checks. For example, we can study image reuse and tamper at different units of figures (compound figures or subfigure) and in specific kinds of figures (Radiology figures or Microscope figures) [11,12]. Beyond graphical integrity issues, our results could be parts of other kinds of chart analysis, such as the aspect of distortion [22] and clearness [59]. For example, our method can be a part of studies of the quality of visual communications from the perspective of human perception and learning [60].

Our future work would benefit from higher-quality images. According to our error analysis, a large proportion of our method's mistakes is due to low-quality images. PubMed Open Access subset makes the article's figures available as separate images, but they are sometimes of lower quality compared to those inside the PDFs. Therefore, future work will attempt to extract images directly from PDFs. Lastly, improvements in these image analysis steps are an active area of research, and therefore should be improved in the future.

In sum, research communities could enhance their education about graphical integrity to all researchers to reduce the burden of publishers and reviewers on graphical integrity. These ideas are already part of research integrity initiatives around the world, including COPE practices and the Hong Kong Principles [61,62].

## 6. Conclusion

This study presents a deep learning-based method to detect graphical integrity issues and research findings of graphical integrity issues in science. This method enables researchers to detect proportional ink violation at scale and offers the potential to address other graphical integrity issues. With this method, we found that graphical integrity issues do appear among scientific articles. Even though the fraction of bar charts having proportional ink violations is

around 5%, one of these violations might cause some readers to misunderstand the research finding because open access journals are easy to access to a wide range of readers. These misunderstandings might harm readers extensively when the research finding is related to safety or health. With these potential consequences, we suggest publishers and research communities consider taking actions to protect readers from graphical integrity issues in publications. In the future, we hope to incorporate this analysis with other kinds of graph checks and improve the accuracy of the components in our method.

## Supporting information

**S1 Table. Summary of Annotation pipeline and Prediction pipeline.** With annotation pipeline, we applied compound figure classification, subfigure separation, and bar chart classification to obtain bar charts from this sample and then ask annotators to annotate graphical integrity issues on these bar chart. With prediction pipeline, we applied our whole graphical integrity issues detector on this sample. Both sets are similar, as demonstrated by analysis in Fig 2.
(XLSX)

**S2 Table. Regression table.** This table shows our analysis of the relationship between proportional ink violations and a group of variables (journal rank, research field, researcher seniority, affiliation country, and year of publication).
(XLSX)

**S1 Fig. Example of graphs with graphical integrity issue.** If the y-axis does not start from zero(as upper two graphs) or there is partially hidden(as lower two graphs), then the bar chart would be labeled as "inappropriate".
(EPS)

**S2 Fig. Text Localization (figures on the left) and Text Role Classification (figures on the right).** We first used a convolutional neural network (YOLO v4, pre-trained on MS COCO dataset) to localize texts on figures. Then, using text role classification to predict the role of texts for feature engineering.
(EPS)

**S3 Fig. Precision, Recall, F1 score, and the ratio of graphical integrity issues under different thresholds.** We could change the threshold to increase our predicted prevalence, but we do not necessarily need to approximate human-annotated prevalence.
(EPS)

**S4 Fig. Example of False Positive cases, which means our method predicted these graphs have graphical integrity issues but actually not.**
(EPS)

**S5 Fig. Example of False Negative cases, which means our method predicted these graphs don't have graphical integrity issues but actually do.**
(EPS)

**S1 Text. Statistical analysis for annotation pipeline.**
(DOCX)

## Author Contributions

**Conceptualization:** Han Zhuang, Daniel E. Acuna.

**Data curation:** Tzu-Yang Huang.

**Formal analysis:** Han Zhuang, Daniel E. Acuna.

**Funding acquisition:** Daniel E. Acuna.

**Investigation:** Han Zhuang, Daniel E. Acuna.

**Methodology:** Han Zhuang, Daniel E. Acuna.

**Project administration:** Daniel E. Acuna.

**Resources:** Han Zhuang, Daniel E. Acuna.

**Software:** Han Zhuang, Tzu-Yang Huang.

**Supervision:** Daniel E. Acuna.

**Validation:** Han Zhuang, Tzu-Yang Huang.

**Visualization:** Han Zhuang, Tzu-Yang Huang.

**Writing – original draft:** Han Zhuang, Daniel E. Acuna.

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
