## [Decision Letter · Decision Letter 0]

16 Aug 2021

Dear Dr. Acuna,

Thank you very much for submitting your manuscript "Graphical integrity issues in open access publications: detection and patterns of proportional ink violations" for consideration at PLOS Computational Biology.

As with all papers reviewed by the journal, your manuscript was reviewed by members of the editorial board and by several independent reviewers. In light of the reviews (below this email), we would like to invite the resubmission of a significantly-revised version that takes into account the reviewers' comments.

As you can see from the reviewers' critiques, there is interest in your approach, and at the same time there are substantial recommendations for clarification, extension, illustration with examples, and more. We look forward to a substantially revised manuscript that addresses these critiques, and provides access to the code so that the reviewers can better assess the computational approach used. 

We cannot make any decision about publication until we have seen the revised manuscript and your response to the reviewers' comments. Your revised manuscript is also likely to be sent to reviewers for further evaluation.

Sincerely,

Feilim Mac Gabhann, Ph.D.

Editor-in-Chief

PLOS Computational Biology

Reviewer's Responses to Questions

**Comments to the Authors:**

Reviewer #1: PLOS Computational Biology

Graphical integrity issues in open access publications: detection and patterns of proportional ink violations

In this paper, Han Zhuang et al. develop a deep-learning based process to measure if bar graphs violate the "proportional ink principle", i.e. where the y-axis does not start at 0. On a dataset of 300K open access figures, such violations were found to be associated with certain journal types and countries.

This paper is an important step in providing better image preparation guidelines and insights into the prevalence of misleading data presentation. However, the description was not always clear, and I had trouble following what the paper was trying to describe. In particular, the introduction and the description of the two datasets could be more clear. Hopefully the authors can address the issues below to make the study easier to understand for a wide audience.

General comments

1. For readers who are not familiar with the "proportional ink principle" it might be hard to understand what the study was about, by just reading the title, abstract, and introduction. The paper itself only defines proportional ink violation at the bottom of page 11, and I had to look it up online in order to understand the paper. It is not a difficult concept, though, just something that could easily be explained in one or two sentences in the Introduction. Page 2 already more or less gives the definition in 'For example, truncating the y-axis of bar charts ', but there it was not clear that this refers to proportional ink. This would be a perfect spot to introduce the principle of proportional ink.

2. Did this paper only look at bar graphs? There is a lot of text on classification but I was not sure which types of images were eventually analyzed. It would be very helpful if the Abstract could contain a short definition of proportional ink, and specifically mention that this paper focused on bar charts (assuming this is the case).

3. The paper describes two study sets of images (see e.g. Table 1) but it was unclear how they differed. They are not clearly described in the Methods. Are both sets subsamples from PMC? Figure 2 shows the two sets are nearly identical but it is unclear how they are different. Confusingly, the 50K set is labeled as "human annotated" in Table 1 while the manuscript appears to describe that these images were processed by automated tools. The authors mention "traditional steps of chart mining" at the bottom of page 4, but this was very vague. Did the authors manually identify the 8K bar charts from the 50K figures or was this done with some automated extraction? The second set "Predictions" is even less well defined. Is this the set that the authors applied their novel analysis tool on? See additional questions about Table 1 below.

4. The introduction might be more clear if the authors could mention that Figures in papers can consist of many elements, including photos and line graphs, and that integrity issues can be found in all kinds of image types.

5. The Introduction appears to state that proportional ink violations amount to research misconduct, but that seemed a bit too harsh. I could see someone leaving out the 0 on a bar graph, without the intention to mislead (maybe with good intentions even) but that would not necessarily qualify as research misconduct. There are many sorts of questionable research practices, but it appears too black and white to assume that a violation of proportional ink is always research misconduct. The introduction could benefit from describing the full spectrum of the Perfect Honest Study to questionable research practices to science misconduct.

6. An additional part of the paper that needs to be better defined is the evaluation of estimating integrity issues. The authors define precision and recall, but they do not define what was compared to what. I assume they compared some automated technique with some manual assessment, but I could not find this in the paper. Is this done on the 50K or on the 250K set?

7. Maybe I missed it but where is "our algorithm" provided? Is the method given by the authors or described in some reproducible method section? Figure 7 appears too general for others to replicate the method.

8. The authors sometimes claim their new method to determine violation of the proportional ink principle to be highly accurate (e.g. page 16), but on page 8 they write "We found that the percentage of graphical integrity issues is significantly larger in human-annotated than automated prediction bar charts". This might be my poor understanding of what actually is compared here, but are these two sentences not in conflict with each other?

Specific comments

9. Page 1, Abstract. Suggested rewording: "However, the extent of violations of these principles in academic publications is unknown."

10. Page 1, Introduction. "Intentionally or unintentionally". Isn't research misconduct not usually done intentionally?

11. Page 1. 'A survey of researchers revealed': the authors cite an older study. Are there any more recent papers? Two recent preprints on this topic could be added: https://osf.io/preprints/metaarxiv/vk9yt/ and https://osf.io/preprints/metaarxiv/xsn94/

12. Page 2, 'For example, truncating the y-axis of bar charts ' would be a perfect spot to introduce the principle of proportional ink.

13. Page 2. 'For example, researchers have developed computer techniques to detect image tampering and image reuse' - this is focusing more on photographic images, correct? Or were graphs also included? Might need to clarify here that these were done on photos, and that this paper will look at graphs.

14. Page 2. 'These studies reveal the prevalence of image fabrications in science beyond small samples of images (Bik et al., 2016a).' That study was a manual analysis of photographic images, not an automated technique applied to all image types.

15. Page 3: 'Data extraction. It is stated that 'charts are images without data' but the text then continues to extract the data from the images. This was a bit confusing; charts contain a lot of data. Do the authors mean that those graphs do not contain actual numbers?

16. Page 4. Chapter 2.3. Typo. Title of paragraph should be "Chart mining". Also, check sentence 'significant challenges left when deadline with low resolution images' - should this be 'dealing'?

17. Page 4. Chapter 2.2 "Datasets about science". Should this not be 2.4? Also, 'In the recent past, we lacked the datasets necessary to analyze science.' - do the authors mean 'to analyze the scientific literature'?

18. Page 4, Chapter 3.1.1/Table 1. It might be helpful to refer to Table 1 here. What is meant by "We labeled images" - do the authors mean "we analyzed'? And by "images" do the authors mean figures or figure panels? What is meant by "we perform the traditional steps of chart mining"? Is this a manual screen of the 50,000 figures, as suggested by "human annotations" in Table 1? Or were these images extracted with some automated tool? This was very confusing.

19. Page 5, Chapter 3.1.2. I did not understand the AUC as a performance measure. How did the authors determine true and false positivity? What is the gold standard here? Is that done on the "Predictions" Dataset? Against what is this measured? Is this a comparison of a manual analysis vs a computer analysis? I was completely lost here.

20. Page 8. Chapter 3.2.2. 'Our algorithm' - not defined. Is that one that can recognize and extract barcharts starting with a large amount of PMC images or from pdfs? Or is it an algorithm that works on human-selected bar graphs to determine if e.g. the y axis starts with 0 or not? How did a human extract bar graphs then? Typo: "extend".

21. Page 8, section 3.2.3. Check sentence: 'higher the ranking (here, “higher” means closer to 1), the more likely to have articles with graphical integrity issues'

22. Page 9, figure 3 - I had trouble interpreting this figure. Do the dots at the top of the graph mean that 100% of papers in certain journals have proportional ink issues? Also, the linear regression line, in particular the rise on the left hand side, appears to be a bit of a stretch. Does that mean that high impact journals have a higher chance of including articles with graphical integrity issues? That is different than the hypothesis stated in 3.2.3. Also typo in figure: "artclles'. The title of the graph also appears to be very general " graphical integrity issues" could be more specific, i.e. proportional integrity issues with bar charts. How many papers/images were included in this graph?

23. Page 9, figure 4. How many papers/images were included in this graph?

24. Page 10, Chapter 3.2.5. How did the authors determine the country of a paper? Did they include all affiliations, or only those of the corresponding author?

25. Page 11, section 4.1.1. 'mentioned in the previous subsection' - should this not be 'mentioned in the following subsection'?

26. Page 12, Chapter 4.1.3. "See the summary of training data and validation data in 2" - should this be Figure 2? Table 2?

27. Page 12, Impact of journals. How was the journal rank determined? I imagine there are multiple indexes to calculate this - which one was used? Or was impact factor used to determine rank?

28. Page 13, Research fields. "we collected the field of study of journals of articles in our sample." - how was this determined? Which fields from which databases were used? Or was this manually determined? More details are needed here.

29. Page 13 'After we have compound figures'  'In the case of a compound figure'

30. References. Some papers appear to be included twice. Examples: Bergstrom C et al. 2020a/2020b; Bik et al. 2016a/2016b; Lee P 2018.

Reviewer #2: The paper investigates violations of the principle of proportional ink in a large open access repository of science papers. The authors find a small but significant proportion of papers that violate this visual rule of data presentation after applying an automated detection method. The work adds to growing body of literature detecting fraud and other forms of misinformation in the scientific literature. The question and results are novel and relevant to the broader scientific community, publishers, and policy makers. However, there are some elements of the analysis and presentation of the results that need to be addressed. I note those issues starting in the third paragraph below.

There are many ways that data graphics are manipulated (sometimes nefariously but most often by mistake) to tell different stories. This paper focuses on the principle of proportional ink, which states that the ink devoted to the representation of numbers should be proportional to those numbers. In particular, this paper focuses on violations of this principle as it applies to bar charts. This is a good place to start given the commonness of this form of data presentation and the relatively straightforward way of testing violations of this graphic. The authors of this study look for truncations of the Y-axis of bar charts as the key indicator of this violation. Overall, they find through hand labeling about 5% of papers and bar charts violate the rule; they find a smaller number (~3%) when conducting the analysis through automation. They then ask questions regarding region, journal, author, etc., which are some of the most interesting aspects of the paper.

The authors report graphical integrity issues in 6.5% of the publications with bar charts and 5.5% per bar chart when hand labeling. They then see a drop after the automated detection (3.6% and 2.3% respectively). What might explain this drop? What could be done with the methods to improve this? The authors noted this underestimation, but I think it would improve the integrity of the results if this was further investigated. I wonder if there is something fairly straightforward that could be addressed that would either explain the gap or reduce the gap. Since the results are cut nearly 50% when automating, it indicates a potentially serious issue when moving to automation. This needs to be addressed in order to improve confidence in the automated detection.

The core contribution relates to the principle of proportional ink. This was a term that the calling bullshit project wrote about in 2017 [Bergstrom, Carl, and Jevin West. "The Principle of Proportional Ink." URL: https://callingbullshit. org/tools/tools proportional ink. html (2017)]. The authors cite Tufte, which is correct for the more general principle of data ink, but Tufte did not come up with the principle of proportional ink. The authors cite Tufte in the first paragraph of their literature review in their description of the principle. Although the principle of proportional ink is inspired by Tufte’s, it is a special case of his more general principle that Tufte talks about.

More follow-up on the interpretation of the results would further strengthen the paper. For example, how would the results differ form the non-open access literature? What are aspects of the bar charts could be examined other than just the Y-axis (e.g., width of bar chart)? When reporting results, how much to the violations break the rule — not just whether they break it or not? There are likely some violations that are minor versus ones that are egregious. What explains some of the country wide and journal results? Why is computer science possibly the worst in these violations? Is something about their presentation of results? If authors are not repeating errors, what does this mean? In the conclusion, the authors mention that a minor problem. But is it minor? 1 out of every 20 articles has a problem. That may be more of a problem (or maybe not). A more thorough discussion here would add to the richness of the study. A super interesting result is that higher prestige journals have relatively higher violation rates. Why is that?

I know that the authors don’t imply that these errors are solely a result fabrication and purposeful manipulation. I imagine that most of these are honest mistakes and the authors note this in the third sentence of the paper with “intentionally or unintentionally” but then continue the sentence with “fabrication, falsification, or plagiarism.” Each of those seems to have an intention to deceive. I recommend rewording to better distinguish the unintentional mistakes and the intestinal falsifications. I just don’t want this method to be confused with the ability to detect intention.

There is potential for publishers to use these methods. This would be a good first line of defense for detecting these kinds of problems before papers are published, but what are some things that publishers should consider if they were to use these methods? Given the policy implications of this work, it would be helpful to go in more depth on the implications of this work and the ways in which the methods could be misused. If done well, it improves the chances of this work to be put in to practice. I don’t think this requires a whole new paper but possibly a section devoted to this.

The presentation needs improvement. For example, the abstract would be improved with more specific statements of the results. The authors state that their “results reveal that graphical integrity issues are significantly more prevalent in some types of journals, research fields, and regions of the globe.” The literature review is sparse and disconnected into a series of bullet-point like lists. Section 2.2, which is an important part of the paper, is presented more as a series of short notes rather than an integrated discussion of the various methods for chart mining. There are missed follow-ups. For example, the authors note that “it is hard to detect and has detrimental effects.” What are the detrimental effects? In presenting the general idea, it would be helpful to provide some specific examples of charts that violate this principle and how it impacts the interpretation of the results. The section 3.1.2 is important for explaining to a more general audience how the results were analyzed. I found the section sparse and difficult to follow. In particular, it is critical to helping the reader interpret results like “0.77 with 0.0209 standard error.” What does this mean for a general reader? Is this good, bad or dependent on other aspects of the problem and results? There are multiple listings of references in the bibliography.

Other presentation issues:

- “estimating the extend of graphical integrity issues”

- “In the future, we hope to incorporate this analysis with other kinds of integrity checks such as”

- Figure 2 title: “therefore validating our prediction step”

- “We then analyzed at the author level.” Weren’t you already doing this?

One the key contributions of this paper is the data set of hand labeled bar charts and automated labeling results and the code for extracting and automating the detection of this rule. They authors note that the data and code will be made available in the PLoS front files, but I did not see it in the main manuscript and could not find the GitHub links. This would be useful for reviewing the methods and results, but at the very least, I hope this is made available upon publication.

Reviewer #3: Dear authors,

Thank you very much for your interesting article "Graphical integrity issues in open access publications: detection and patterns of proportional ink violations". You focus on a topic that has so far received little attention in the research integrity community. Your approach to investigate integrity issues in an automated way using neural networks is an important contribution to our research field and can provide insights that give a more accurate picture of where urgent action is needed to promote scientific integrity.

Several questions arose after reading:

-you address Tufte in the literature review, but not theorists who address aspects of the perception of visual information, such as Bertin, or Ware. At the same time, in 4.2.2 feature 3, you name the distance of the y-axis label from the x-axis as a possible factor that can undermine integrity-isn't that also a perceptual factor? What is your assessment of the importance of perceptual factors?

-You describe the results of the analysis and a conservative estimate of the number of inappropriate graphs in the database in section 3.2.2. It remains unclear though how high the proportion of publications with graph integrity problems is compared to the total of all analyzed publications. This is certainly because you focused exclusively on publications with bar charts, but it would be helpful for readers to get an idea of the extent of the problem.

-In the same section, you emphasize that your method produces fewer results compared to human annotation, thus underestimating the problem. How do you explain this difference? If your classification allows for such inaccuracy, is it really sufficiently effective?

-How do you deal with the fact that bar charts are used very differently in different disciplines? How is that reflected in your statistics?

-After reading the article, it is still not clear to me whether the graphical integrity classification system developed by you allows a sufficient statement about the integrity of a graph, or whether there are — similar to other types of images, or even texts — gray areas that have to be included when deciding about the integrity. So, at what point is a bar chart to be labeled "inappropriate" — how many criteria have to be considered to make a reliable judgment? Possibly some concrete examples could provide clarity here and strengthen your argumentation?

-in section 3.2.4, you note that there are significant differences between academic disciplines when it comes to graphical integrity. Have you explored whether these differences may also be because there are guidelines for handling images and graphs in certain disciplines and in others not?

-in section 3.2.5, it remains unclear what criteria you used exactly to undertake the investigation "by country". When you compare „countries“, does that mean you compare the publications of journals published in a given country, or is it about the origin of the authors? Or something completely different?

General comment: there are a number of spelling and typing errors that leave the notion the paper was written under a certain time pressure. These should be corrected before publication.

Subject-specific abbreviations should be explained to facilitate understanding (e.g. p. 5, item 3.1.2 ROC / AUC etc.).

Under Data and Code Availability it is implied that all data and code will be published via GitHub and Zenodo. However, I could not find a corresponding link.

Hope you find some of these comments helfpul, again thank you for your work!

**Have the authors made all data and (if applicable) computational code underlying the findings in their manuscript fully available?**

Reviewer #1: **No: **There is no source code to the algorithm used to classify the images - only a very general flow chart.

Reviewer #2: **No: **The authors say they will make this available but I did not find the data and code repository

Reviewer #3: Yes

PLOS authors have the option to publish the peer review history of their article (what does this mean?). If published, this will include your full peer review and any attached files.

Reviewer #1: **Yes: **Elisabeth Bik

Reviewer #2: No

Reviewer #3: No
---

## [Decision Letter · Decision Letter 1]

10 Nov 2021

Dear Dr. Acuna,

Thank you very much for submitting your manuscript "Graphical integrity issues in open access publications: detection and patterns of proportional ink violations" for consideration at PLOS Computational Biology. As with all papers reviewed by the journal, your manuscript was reviewed by members of the editorial board and by several independent reviewers. The reviewers appreciated the attention to an important topic. Based on the reviews, we are likely to accept this manuscript for publication, providing that you modify the manuscript according to the review recommendations. 

Sincerely,

Feilim Mac Gabhann, Ph.D.

Editor-in-Chief

PLOS Computational Biology

[LINK]

Reviewer's Responses to Questions

**Comments to the Authors:**

Reviewer #1: I thank the authors for incorporating the reviewers' comments and for revising the manuscript. It is now much easier to read and understand. In particular, I found the new Supplementary Figure 1 very helpful, and I was wondering if it could even replace Table 1 (with some addition of e.g. the number of papers provided in the Table but not in Fig S1 - decision up to the authors. However, I would recommend to switch the left and right arms of Suppl Figure 1 so that it matches the order of Table 1, and the order in which the analyses are described.

There are still a lot of typos in this version, all of which could have been caught by a simple spell checker, and all of which should be easy to address, without the need of another round of review. I have listed the ones I caught below.

In addition, several of the papers cited in the text are not listed in the References, and I would ask the authors to do a careful check. I have listed the ones I found below.

1. The following papers have been cited in the text but appear to be missing from the References list at the end:

○ Gopalakrishna et al., 2021

○ Bergstrom and West, 2020 (the References lists a 2021 article, though, maybe wrong year?)

○ Powell, 2012

○ Weber-Wulff, 2014

○ Foltýnek et al., 2020

○ Bik et al., 2016

○ Tufte, 2001

○ Bertin, 2011

○ Ware, 2004

○ Davila et al., 2021

○ DarrinEide, 2021

○ Jambor et al., 2021

2. Page 1, Author summary: 'extend' should be 'extent'

3. Page 1, Introduction. 'Inacuracies' should read 'inaccuracies'

4. Page 2: 'quantiative' should read 'quantitative'

5. Page 2: "Researchers have proposed automated and scalable techniques to detect violations of research integrity." still too strong and too focused on integrity issues. Maybe reword to "Researchers have proposed automated and scalable techniques to detect violations of research integrity and responsible research practices"

6. Page 2: 'regions of interests' - 'regions of interest' reads better.

7. Page 4: In 'Chart extraction', should change text to 'that has enabled'

8. Page 4: In 'Chart classification', 'proportial' should read 'proportional'

9. Page 5: "Our annotation pipeline starts with a randomly sample of 50,000 figures (Table 1)." The text could also refer to Supplementary Figure 1 here. Also, the text should probably read "a random sample of 50,000 figures'.

10. Page 5, bottom. 'We found that 356 figures was classified'  'were classified'

11. Page 6, caption of Figure 1. 'annoted' should be 'annotated'.

12. Page 6, sections 3.1.1 and 3.1.2. Between section 3.1.1 and 3.1.2, there suddenly is an algorithm that gets tested, but there is no description stating how or on what set it was developed. Does the sentence at the end of 3.1.1 describe the development of the algorithm, or was it developed in a previous section? Maybe reword that sentence to "These data was used for the development of a machine learning algorithm." Does section 3.1.2 describe the performance of the 'algorithm' on the manually annotated dataset? Is that what is meant by "detector"? I was a bit confused here. It appears section 3.1.2 was done on the 'human annotated' set, but maybe it would be more clear if that was added to the text.

13. Page 6, end of 3.1.2. 'while being relatively less successful with false negative '  'false negatives' (not sure, but please check).

14. Page 7. Caption of section 3.2.1. 'prevalance' should be 'prevalence'

15. Page 8, section 3.2.1. "we have this high false positive" - This sentence did not read well - should this not be "false negative rate in the algorithm-analyzed set"? The previous text suggested that more issues were found in the human annotated set than in the algorithm set.

16. Page 8. In section 3.2.3, 'maximim' should be 'maximum'

17. Page 8. In section 3.2.3, check sentence: "A Pearson correlational analysis revealed that there is no correlation that the higher the ranking" - end of sentence appears to be missing.

18. Page 9, Y axis of Figure 3. 'proportinal' should read 'proportional'

19. Page 10. "Previous research has shown that there are significantly different prevalance of research integrity across countries" - This should be 'prevalence', but should this also read "that there is.."? . It might also be more clear to specifically name the top 3 countries that were found to perform the worst here (although, ouch, the Netherlands?).

20. Page 11. "It is uknown" - typo

21. Page 12. Section 3.2.8. Check sentence 'Interpreting the differences between human annotation and our method improve the transparency of our method but is difficult. '

22. Page 12. Section 3.2.8. Check sentence 'why our annotator find 5.5% of bar charts having proportional ink violation but our method finds 2.3% of bar charts having proportional ink violation'. It would be more clear to change 'why our annotator find' to 'why human annotators found' and to change 'our method' to 'our machine learning method'.

23. Page 12. Section 3.2.8. Check sentence 'We found this discrepancy between predicted prevalence and the human- annotated prevalence can the result of the threshold of classifying a bar chart into proportional ink violation in our classifier'

24. Page 13, section 3.3. ' All the previous analysis are' maybe change to 'All the previous analyses were'

25. Page 13, section 4.1.1. Typo 'classsification'. Change 'collected images mentioned section 3.1.1' to 'collected images mentioned in section 3.1.1'.

26. Page 14. Check sentence: 'Also, there are 4780 authors which we did not find graphical integrity issues in our labeled data.'

27. Page 17, in Figure 8, 'Seperation' should read 'Separation'

28. Page 20. The sentences at the bottom of page 20/top of page 21 should read '...points are images that have a ...' instead of 'images, which'

29. Page 22. ' in an random manner' should read ' in a random manner'

30. Page 22. 'Our study found that a varying prevalence' should read 'Our study found a varying prevalence'

31. Page 22. In the sentence that starts with 'More broadly', 'have' should read 'has'. In the next sentence, 'chance of improve' should read 'chance to improve'.

32. Page 22. In section 5.3, insert a space in 'The method dependson'

33. Page 22. Check: 'Our analysis focuses on violations of principle of proportional ink'

34. Page 23. 'adherance' should be 'adherence'.

35. Page 23. Check 'can have a different review processes'

36. Page 23. 'constainly' should read 'constantly'

37. Page 23. 'stiffle' should be 'stifle'

Signed, Elisabeth Bik

Reviewer #2: The authors have thoroughly addressed my comments and questions. I support seeing this manuscript move forward to publication. I hope this work inspires more research automating the detection of graphical mistakes.

Reviewer #3: Again, thank you for your valuable work. One last thing in terms of content: on page 23 you state: "In this sense, we should constantly be paying attention to the bad actors in a game of cat and mouse. According to our analysis, the extend of the problem is manageable (5% have violations) and the intentional violation of graphical integrity rules must be even less common. Therefore, a cost benefit analysis suggests that we should let the method be used widely." which implies the problem of ink violations is relatively harmless, but some paragraphs later in your conclusions you make the point that the misunderstanding of a graph could well harm peoples lives. This produces a tension you should be aware of. Last not least, there are still some linguistic errors in the text that could be fixed relatively easily and it should also be checked for double spaces.

**Have the authors made all data and (if applicable) computational code underlying the findings in their manuscript fully available?**

Reviewer #1: Yes

Reviewer #2: Yes

Reviewer #3: Yes

PLOS authors have the option to publish the peer review history of their article (what does this mean?). If published, this will include your full peer review and any attached files.

Reviewer #1: **Yes: **Elisabeth M Bik

Reviewer #2: No

Reviewer #3: No

Figure Files:

Data Requirements:

Reproducibility:

References:

---

## [Editor Report · Decision Letter 2]

16 Nov 2021

Dear Dr. Acuna,

We are pleased to inform you that your manuscript 'Graphical integrity issues in open access publications: detection and patterns of proportional ink violations' has been provisionally accepted for publication in PLOS Computational Biology.

Best regards,

Feilim Mac Gabhann, Ph.D.

Editor-in-Chief

PLOS Computational Biology

---

## [Editor Report · Acceptance letter]

7 Dec 2021

PCOMPBIOL-D-21-01149R2 

Graphical integrity issues in open access publications: detection and patterns of proportional ink violations

Dear Dr Acuna,

I am pleased to inform you that your manuscript has been formally accepted for publication in PLOS Computational Biology. Your manuscript is now with our production department and you will be notified of the publication date in due course.

With kind regards,

Zsofia Freund
